# Diabetes Control and Clinical Outcomes among Children Attending a Regional Paediatric Diabetes Service in Australia

**DOI:** 10.3390/nu16213779

**Published:** 2024-11-04

**Authors:** Luke Huynh, Michelle Booth, Uchechukwu L. Osuagwu

**Affiliations:** 1Bathurst Rural Clinical School (BRCS), School of Medicine, Western Sydney University, Bathurst, NSW 2795, Australia; lthuynh2891@gmail.com (L.H.); michelle.booth@health.nsw.gov.au (M.B.); 2African Vision Research Institute (AVRI), School of Optometry, University of KwaZulu Natal, Westville, Durban 3629, South Africa

**Keywords:** type 1 diabetes, multidisciplinary team, rural health, mental health, HbA1c, CSII, MDI, CGM, time in range

## Abstract

Australian children with diabetes commonly struggle to achieve optimal glycaemic control, with minimal improvement observed over the past decade. The scarcity of research in the rural and regional Australian context is concerning, given high incidence rates and prominent barriers to healthcare access in these areas. We conducted a retrospective audit of 60 children attending a regional Australian paediatric diabetes service between January 2020 and December 2023. The majority of patients had type 1 diabetes (*n* = 57, 95.0%); approximately equal numbers were managed with continuous subcutaneous insulin infusion (CSII) pumps vs. multiple daily injections (MDIs), whilst 88.3% (*n* = 53) also utilised continuous glucose monitoring (CGM). The mean age at last visit was 14.0 years (SD, 3.4), mean diabetes duration 5.8 years (SD, 4.6), and mean HbA1c level 8.1% (65.3 mmol/mol); only 36.8% achieved the national target of 7.5% (58 mmol/mol). Mean BMI-SDS was 0.8 (SD, 1.0); almost half (*n* = 27, 45.0%) were overweight or obese. Many patients had mental health conditions (31.7%), which were associated with higher hospitalisation rates (*p* = 0.007). The attendance rate was 83.2%, with a mean of 3.3 clinic visits per year (SD, 0.7); higher attendance rates were associated with increased CGM sensor usage (r = 0.395, *p* = 0.007 Overall, the diabetes service performed similarly to other clinics with regards to glycaemic control. Whilst achieving treatment targets and addressing comorbidities remains a challenge, the decent attendance and the high uptake of healthcare technologies is commendable. Further efforts are needed to improve diabetes management for this regional community.

## 1. Introduction

Type 1 diabetes mellitus (T1DM) is a chronic metabolic disease caused by insulin deficiency [1], and is the most common form of diabetes in childhood [2]. It is a major problem for the Australian healthcare system, costing the nation 19,000 years of healthy life in 2023 and AUD 373 million in 2020–2021 [3,4]. Despite this expenditure, large-scale Australian studies conducted in recent years have consistently exposed the concerning state of diabetes control, with minimal improvement observed over the past decade [5,6,7]; most Australian children fail to meet the internationally recognised glycaemic target defined as having an HbA1c below 7.5% (58 mmol/mol) [8,9].

Although significant research has been undertaken to optimise diabetes management, clinical audits of diabetes centres are comparatively less common. Many were conducted several years ago, and do not reflect changes in the global environment such as the COVID-19 pandemic, or increased uptake of technological advancements [10]. One such technology is continuous glucose monitoring (CGM), with only a few Australian studies having examined its use in T1DM clinics [11,12]. Although the American Diabetes Association (ADA) recommends that CGM be offered as soon as possible [13], Australian guidelines currently do not advocate for its routine use [8]. Poor diabetes control has also been linked with mental health issues [14], with some audits highlighting the need for increased psychological care [6,11]. Studies in rural and regional Australia are rare; two audits conducted over ten years ago evaluated models of care in comparison to metropolitan clinics [15,16]. This lack of research is particularly concerning, given that the highest incidence rates of T1DM are reported in regional locations [17] where significant barriers to healthcare access remain [18].

This single-centre, retrospective audit aimed to characterise the children attending a regional Australian paediatric diabetes service. While small-scale surveys have previously been conducted, no comprehensive study of the clinic exists. We investigated attendance rates, anthropometry, clinical history, and treatment outcomes in light of recent changes to the model of care. These findings can then be used to identify areas for improvement, optimise treatment protocols, and enhance patient outcomes in regional communities.

## 2. Materials and Methods

### 2.1. Study Setting

Bathurst Base Hospital is a Level C referral facility with 100 beds located in regional New South Wales [19]. Its paediatric diabetes service has recently benefited from various changes to its model of care. This included access to a larger space (mid-2022) and the temporary recruitment of a social worker (February 2022 to March 2023) to the existing multidisciplinary team (MDT) consisting of a credentialled diabetes educator (CDE), dietician, and general paediatrician. To provide more time during appointments, HbA1c testing was moved to the pathology unit (22 June 2020), to be performed in patients’ own time instead of in clinic. Children with diabetes are scheduled to attend four clinics per year. These clinics are usually held face-to-face, but telehealth is also an option and was frequently utilised during the COVID-19 pandemic. A retrospective audit was conducted of all 60 children who attended the clinic between 1 January 2020 and 31 December 2023.

### 2.2. Clinical Targets

The target levels for diabetes control used in the clinic were set in accordance with national and international guidelines. Optimal glycaemia was defined as having an HbA1c level below 7.5% (58 mmol/mol) [8,9]. Optimal CGM target ranges were defined as having blood glucose levels between 3.9 and 10.0 mmol/L during the day (6:00 a.m.–10:00 p.m.), and between 4.4 and 8.3 mmol/L at night (10:00 p.m.–6:00 a.m.); optimal times in the target range were defined as >70% in range (TIR), <4% below range (TBR), and <25% above range (TAR) [19].

### 2.3. Data Collection

An Excel spreadsheet was used to collect the data, which were securely stored in a OneDrive folder. Types of data collected included patient information, demographics, attendance, reasons for missed appointments, anthropometry, clinical profile, treatment modalities, clinical outcome measures, and reasons for leaving the clinic. Attendance data were obtained from the clinical records kept by administration staff. All other data was obtained from Western New South Wales Local Health District (WNSWLHD) electronic medical records (eMRs) or from specialist letters and reports. This manuscript was produced according to the RECORD statement guidelines [20].

The clinical profile included anthropometry, substance use, diabetes-related hospitalisations, comorbidities, and mental health conditions. Clinical outcome measures included CGM metrics, as well as point-of-care (POC) HbA1c levels, when available. CGM data were obtained for the largest period possible immediately before the final assessment (typically 90 days). Attendance rate was calculated as the number of clinics attended divided by the number of clinical appointments expected during the study period. Anthropometry was taken at the first and last assessment. In Australia, WHO growth charts are used for children aged 0–2 years, while US-CDC growth charts are used for children aged 2–18 years [21,22]. Since the patients were older than two years, the US-CDC cut-offs were used; a BMI for age between the 85th and 94th percentile was considered overweight, and above the 95th percentile was considered obese [23]. Change in HbA1c over time was calculated as last HbA1c minus first HbA1c.

### 2.4. Ethics Approval

The audit was approved as a quality assurance activity on 23 March 2019 by the Western Sydney University Human Research Ethics Committee (2019/ETH04433, 23 March 2019).

### 2.5. Statistical Analysis

All data analyses, encompassing both descriptive and inferential statistics, were conducted using Microsoft Excel 365 (version 2402). Categorical variables were summarised by frequency and percentage, whilst continuous variables were summarised by mean ± standard deviation (SD), or range where appropriate. This audit reported the characteristics of all patients seen at the diabetes clinic within this period for quality improvement. However, for comparing treatment targets with national averages, data for only T1DM patients were used, due to the small number of people with other types of diabetes. Significance was defined as *p* < 0.05 and was calculated using two-tailed *t*-tests and Pearson correlation coefficients.

## 3. Results

### 3.1. Patient Characteristics

Table 1 represents the characteristics of the sample population. All patients were born in Australia and spoke English as their primary language, with a small proportion (*n* = 7, 11.7%) being of Indigenous background. The majority of patients had type 1 diabetes (57, 95.0%). The mean age at diagnosis was 8.5 ± 4.2 years (range, 1–16), with a mean diabetes duration of 5.8 ± 4.6 years.

During the study period, 28 patients were new to the clinic, 19 transitioned to adult care, and 5 left for various reasons. The overall attendance rate was 83.2%, with a mean attendance of 26.3 ± 14.3 months. Common reasons for missing appointments included sickness, family issues, difficulties with uploading insulin pump data, lack of transportation, and poor organisation. Out of the 99 clinics held within the four-year study period, only 1 clinic was cancelled, at the start of the COVID-19 pandemic. The mean number of visits per year was significantly higher (3.3 ± 0.7 vs. 3.1 ± 1.1; *p* = 0.030) than that reported in a national audit [7].

### 3.2. Clinical Profile

Table 2 represents the clinical profile of the sample population. At final assessment, 55.5% of patients were of a healthy weight, 26.7% were overweight, and 18.3% were obese, and none were underweight. Mean BMI-SDS increased between first and last assessments (0.57 ± 1.01 vs. 0.80 ± 0.97; *p* = 0.002), but the final score did not differ from what is expected in children with T1DM (0.80 ± 0.97 vs. 0.87 ± 1.09; *p* = 0.583) [24].

Substance use, including smoking, alcohol, and drugs, was uncommon (*n* = 9, 15.0%) although not rare in this cohort. On the other hand, mental health conditions were common (19, 31.7%), including depression (8, 13.3%), anxiety (10, 16.7%), and ADHD (10, 16.7%); the mean age at last visit was 15.6 ± 6.2 years for this group. Compared with those without, those with mental health conditions had higher HbA1c readings (7.9 ± 1.6% vs. 8.6 ± 2.5%) which was almost significant (*p* = 0.059). Having a mental health condition was also not associated with attendance rates (*p* = 0.734) or CGM usage (*p* = 0.358).

There were 47 hospitalisations during the study period (four years), involving over one-third of this cohort (21, 35.0%) including 16.7% with DKA, 18.3% with hyperglycaemia, and 20.0% for elective admissions. Patients with mental health conditions had significantly higher hospitalisation rates than those without (*p* = 0.007), and were disproportionately represented in the number of hospitalisations (61.7%). Number of hospitalisations was also associated with higher mean HbA1c levels (*r* = 0.498, *p* < 0.001). 

### 3.3. Diabetes Management and Clinical Outcomes

Table 3 represents the diabetes management modalities and clinical outcomes of the sample population. Approximately the same number of patients were managed with continuous subcutaneous insulin infusion (CSII) pumps vs. multiple daily injections (MDIs), but one older patient with T2DM was no longer treated with insulin. Most (*n* = 53, 88.3%) had access to continuous glucose monitoring (CGM), but full data were only available for 84.9% (*n* = 45/53). There were no significant differences between CSII and MDI with regard to patient outcomes.

The mean time spent in the target range was 57.5%, with only 17.8% (*n* = 8/45) of patients meeting the clinical target (TIR > 70%, TBR < 4%, TAR < 25%). CGM sensor usage varied greatly between patients, with a mean of 82.5 ± 24.6%. Higher sensor usage was associated with higher TIR (*p* = 0.035), higher attendance rates (*p* = 0.007), and lower HbA1c (*p* = 0.022).

HbA1c data were available for 98.3% of patients; there was no significant change between first and last assessments, overall (8.5 ± 2.0% vs. 8.0 ± 1.9%; *p* = 0.085). The mean HbA1c for T1DM patients (8.1 ± 1.4%) was similar to that reported in a large-scale multi-centre national audit (8.3 ± 3.5%) [7], an international study (8.2 ± 1.4%) [25], and a single-centre regional Australian study with a similar population (8.1 ± 1.3%) [16]; nevertheless, it was higher than the national target of 7.5%, which was achieved by only 36.8% of T1DM patients (*n* = 21/57) in this study. 

### 3.4. The Paediatric Diabetes Clinic

Unlike in metropolitan hospitals, the paediatric diabetes clinic is not a separate specialist service, but is incorporated into the Bathurst Hospital diabetes service. Currently, 36 children with diabetes attend the clinic. Two general paediatricians oversee approximately half of the children each, in addition to their regular hospital duties. The service’s multidisciplinary team (MDT) is not exclusive to paediatric patients, but provides care for all patients with any form of diabetes. It includes a full-time equivalent (FTE) of 1.0 dietician, 1.5 credentialled diabetes educator (CDE), and no social worker or psychologist.

## 4. Discussion

In this first comprehensive audit of this regional paediatric diabetes service, we have demonstrated suboptimal control of T1DM which falls short of the national target. This has occurred despite better-than-expected attendance rates and extensive uptake of continuous glucose monitoring (CGM) in all patients. CGM sensor usage varied greatly between patients, and was associated with higher clinic attendance rates and improved glycaemic outcomes. Almost half of this cohort were overweight or obese, with the proportion increasing slightly between the first and last assessments. Mental health conditions and diabetes-related hospitalisations were also common in this population. Those with a mental health condition were more likely to experience higher hospitalisation rates.

The findings of this study reflect an overall global trend whereby only about 15.7–46.4% of children with T1DM meet their HbA1c targets [26]. However, it is noteworthy that a greater proportion of the T1DM patients in this study (36.8%) achieved the HbA1c target compared to those in the national audit (27%) [7], a large-scale, multi-centre Australasian study (27.2%) [6], and a UK national audit (14.7%) [27]. Despite this, the mean HbA1c level (8.1%) was similar to the national (8.3%) [7] or international (8.2%) averages [25], suggesting that, while the clinic has been relatively successful in helping individual patients, overall management remains a challenge. Addressing these issues early is crucial, given that failing to meet glycaemic targets is strongly linked with the development of diabetes complications later in life [28]. Further efforts are needed to clarify the factors causing some patients with T1DM in this regional community to fall behind.

Our results align with findings from Goss et al. [16], who implemented a multidisciplinary model of care called ‘RADICAL’ in a similar population [16]. In addition to a significant reduction in mean HbA1c from 9.6% to 8.1%, their study reported improved quality of life for rural youth with T1DM, eliminating the gap previously seen between rural and urban diabetic populations. The diabetes clinic is already making good progress with a multidisciplinary approach; thus, continuing to strengthen this model of care could further improve patient outcomes.

Despite staff concerns for the contrary, clinic attendance rates were surprisingly good (83.2% overall). The mean number of clinic visits (3.3 visits) per year was higher than the 3.1 reported in a national audit in 2017 [7], but lower than the 3.7 reported in an earlier audit in 2010 [6]. The 2010 audit was conducted across multiple centres that offered appointments only three times per year (three centres), or four or more times per year (ten centres). Thus, this number should only be considered in absolute terms, rather than as a measure to compare overall attendance rates. The large difference between the two previous studies could be attributed to a difference in the study populations—the 2017 audit involved the five largest paediatric diabetes centres in Australia, whilst the 2010 audit involved eighteen Australasian centres with no size criteria. An important consideration is that our study time frame encompassed the duration and aftermath of the COVID-19 pandemic, a period that saw the rapid adoption of telemedicine. This increased uptake made diabetes telehealth consultations the third-highest in Australia [29], and may have contributed to the differences in attendance rates.

Lack of transportation, one of the causes of missed appointments, is known to be a major barrier to healthcare access in regional Australia [18,30]. However, technical issues with insulin pumps also led to the rescheduling of many appointments. This is concerning, as multiple studies have shown that confusion with medical technology impacts therapeutic effectiveness [31,32]. The clinic addressed this by providing education sessions for patients led by representatives from insulin pump manufacturers, which has been well received by patients’ families. Providing further educational activities regarding pump use and diet control outside of the clinic could also be beneficial.

There was a high prevalence of overweight and obesity in this clinic population (45% by final assessment); this was higher than the rate reported in the national audit (33%) [7] and in the general population (27.7%) [33]. It has been established that children in regional areas are more likely to be overweight compared to those living in major cities [34]. In T1DM children, higher BMI has been associated with a higher incidence of diabetes-related complications and comorbidities, and should be addressed as early as possible [35]. This viewpoint is reflected in current Australian guidelines, which advocate for dieticians as integral members of the MDT [8]. The International Society for Paediatric and Adolescent Diabetes (ISPAD) recommends 0.5 FTE dieticians per 100 patients [36], which is higher than the 2010 Australian average of 0.19 per 100 patients [6]. The Bathurst diabetes service currently employs a 1.0 FTE dietician; however, as the dietician is not exclusively involved in paediatric diabetes care, it is difficult to determine whether this number is sufficient for the clinic.

This cohort exhibited significantly higher rates of mental health disorders (31.7%) compared to the general population—14% of children aged 4–11 years [37] and 14% of adolescents aged 10–19 years [38]. Patients with mental health disorders (26.7%) were also disproportionately represented in the number of diabetes-related hospitalisations (61.7%). Although mental health was only associated with some patient outcomes, this does not rule out the impact of mental health on diabetes care. A similar study of a transition clinic demonstrated significant associations between mental disorder and diabetes complications [11]. Our small sample size may have limited our ability to detect small changes and, while HbA1c is a key metric, it cannot provide a comprehensive picture of diabetes management for a patient. Nevertheless, the importance of providing psychosocial support for adolescents with diabetes is well-established [8]. The mean age of those with mental health conditions was 15.6 years at final assessment, reflecting the increased risk of psychiatric comorbidity associated with longer diabetes duration [39]. Despite the majority of patients being young adolescents, the current MDT lacks a social worker or psychologist; according to the ISPAD guidelines, there should be one 0.3 FTE social worker/psychologist per 100 patients [36].

The proportion of patients managed with CSII and MDI were higher, in each case, than national rates (53.5% and 45.0% vs. 44.0% and 38.0%, respectively) [7], as none were managed with bidaily injections. Although current evidence suggests that there are small advantages to using CSII over MDI on average, no definitive conclusions can be drawn [8]. Our findings reflect this uncertainty, showing no clear benefit of one treatment modality over the other. Nevertheless, the CSII uptake in this clinic far surpasses the rates 10 years ago, when only 12% of children with T1DM had access [40]. Our study could not evaluate the effectiveness of CGM compared to self-monitoring, due to the large CGM uptake in the clinic (88.3%). However, we demonstrated that increased sensor usage was associated with improved clinical outcomes. These results align with findings from a recent study of 25,383 children with T1DM, which reported that CGM use was associated with meeting HbA1c targets [41]

Our study is the first to characterise the children attending this regional clinic, and is likely the first of its kind conducted in regional Australia post pandemic. The main strengths were a moderately longer time frame compared to other single-centre audits, and a comprehensive data set allowing for thorough analysis. However, the study’s retrospective design, lack of control variables, and small sample size limit its statistical power. Some patient records were difficult to access on the eMR; a few HbA1c levels were not recorded on the database and could not be included in the study. We did not account for differences in appointment durations between paediatricians, the effects of changes to the model of care, or the impact of socioeconomic factors. Despite this, our focus on real-world outcomes like mental health, BMI, hospitalisation, and healthcare barriers contributes to a nuanced understanding of paediatric diabetes care in a regional setting. Future studies should include a large qualitative aspect, to assess patient perceptions of diabetes healthcare practices. Mental health conditions could be further categorised for a more nuanced understanding of their impact on diabetes control. A longer study time frame, multiple centres, and broader scope of variables (complications, screening, and metabolic outcomes) would enable a more detailed examination of factors associated with achieving target outcomes. New technologies, such as the hybrid closed-loop system, or “artificial pancreas”, are poised to revolutionize T1DM care; future studies should explore their integration into wider clinical practice. In addition, although the use of data from a single centre limits the generalizability of the results, the present findings offer relevant insights for other clinics in rural and regional settings. Similar glycaemic control outcomes, despite resource limitations, suggest that key strategies, such as promoting high attendance and CGM usage, can be applied broadly to improve patient engagement and outcomes. Additionally, the observed impact of mental health comorbidities on hospitalisation highlights the importance of integrated-care approaches, which could benefit clinics managing similar patient needs. These results support adaptable strategies for improving paediatric diabetes care across varied healthcare environments.

## 5. Conclusions

This was the first comprehensive study of the children attending Bathurst’s diabetes service. We evaluated attendance rates, anthropometry, clinical history, and treatment outcomes, in light of recent changes to the model of care. Our findings shed light on the complex landscape of diabetes management in regional Australia. Despite various changes to the model of care, achieving optimal glycaemic control and addressing comorbidities remains a challenge. The high prevalence of mental health conditions among patients highlights the importance of integrated psychosocial support within care teams. Although there are many areas for improvement, the clinic has performed well with regard to attendance and uptake of healthcare technologies. By leveraging these findings, clinic staff can work towards more effective, patient-centered approaches that optimise outcomes and quality of life for children with type 1 diabetes in regional communities.

## Figures and Tables

**Table 1 nutrients-16-03779-t001:** Demographics, diabetes history, and attendance records of the sample population attending the paediatric diabetes clinic.

Variables	All
Demographics	
Male, *n* (%)	25 (41.7)
Female	35 (58.3)
Place of birth—Australia	60 (100.0)
Language—English	60 (100.0)
Religion—Christian denominations	31 (51.7)
Religion—none	29 (48.3)
Indigenous	7 (11.7)
Current age, mean ± SD	14.9 ± 4.0
Diabetes history	
Type 1 diabetes, *n* (%)	57 (95.0)
Type 2 diabetes	2 (3.3)
MODY	1 (1.7)
Age at diagnosis (years), mean ± SD	8.5 ± 4.2
Diabetes duration	5.8 ± 4.6
Clinic attendance	
Age at entry (years), mean ± SD	11.8 ± 3.7
Age at last visit (years)	14.0 ± 3.4
New to clinic at entry, *n* (%)	28 (46.7)
Left clinic—transitioned	19 (31.7)
Left clinic—other ^1^	5 (8.3)
Months in clinic	26.3 ± 14.2
Mean visits per year, mean ± SD	3.3 ± 0.7
Number of missed clinics	1.9 ± 1.8
Attendance rate (%)	83.2 ± 16.1

Abbreviations: MODY = maturity-onset diabetes of the young. ^1^ Three moved away, one lost contact, one referred to gestational diabetes clinic.

**Table 2 nutrients-16-03779-t002:** Anthropometry, clinical history, and diabetes-related hospitalisations of the sample population attending the paediatric diabetes clinic.

Variables	All
Anthropometry at initial assessment	
Height (cm), mean ± SD	151.8 ± 20.5
Weight (kg)	50.9 ± 19.9
BMI-SDS	0.6 ± 1.0
Normal weight ^1^, *n* (%)	38 (63.3)
Overweight ^1^	16 (26.7)
Obese ^1^	6 (10.0)
Anthropometry at final assessment	
Height (cm), mean ± SD	160.5 ± 15.8
Weight (kg)	62.2 ± 20.2
BMI-SDS	0.8 ± 1.0
Normal weight ^1^, *n* (%)	33 (55.5)
Overweight ^1^	16 (26.7)
Obese ^1^	11 (18.3)
Clinical history, *n* (%)	
Any substance use	9 (15.0)
Smoking	7 (11.7)
Alcohol	7 (11.7)
Drugs	4 (6.7)
Non-psychological comorbidity ^2^	21 (35.0)
Allergies	10 (16.7)
Any mental condition	19 (31.7)
Depression	8 (13.3)
Anxiety	10 (16.7)
Suicidal ideation	7 (11.7)
ADHD	10 (16.7)
ASD	4 (6.7)
Other ^3^	12 (20.0)
Diabetes-related hospitalisations, *n* (%)	
Any hospital presentation	21 (35.0)
DKA	10 (16.7)
Multiple DKA	5 (8.4)
Hyperglycaemia ^4^	11 (18.3)
Multiple hyperglycaemia ^4^	2 (3.3)
Hypoglycaemia	1 (1.7)
Planned admission ^5^	12 (20.0)
Multiple planned admissions	3 (5.0)

Abbreviations: BMI = body mass index, SDS = standard deviation score, ADHD = attention deficit hyperactivity disorder, ASD = autism spectrum disorder, DKA = diabetic ketoacidosis. ^1^ Defined according to US-CDC growth charts; ^2^ including six with thyroid disease and five with coeliac disease; ^3^ including BPD, ODD, PTSD, learning/language/mood/eating disorders; ^4^ hyperglycaemia without ketoacidosis; ^5^ for monitoring or for optimisation of treatment.

**Table 3 nutrients-16-03779-t003:** Diabetes management, CGM metrics, time in range, and HbA1c of the sample population attending the paediatric diabetes clinic.

Variables	All
Diabetes management, *n* (%)	
CSII	32 (53.3)
MDIs	27 (45.0)
Oral ^1^	6 (10.0)
CGM	53 (88.3)
Self-monitoring ^2^	7 (11.7)
CGM metrics	
Uploaded data available ^3^, *n* (%)	45 (84.9)
Sensor usage ^4^ (%), mean ± SD	82.5 ± 24.6
Blood glucose level (mmol/L)	10.8 ± 2.7
Standard deviation	4.0 ± 1.4
Achieved TIR target ^5^, *n* (%)	8 (17.8)
Time in range (%), mean ± SD	
Very low	0.5 ± 0.5
Low	1.8 ± 1.4
In target range	57.5 ± 19.7
High	22.6 ± 7.6
Very high	19.4 ± 16.0
Overall HbA1c (%, mmol/mol)	
*n* (%)	59 (98.3)
Mean, mean ± SD	8.1 ± 1.4 (65.1 ± 15.5)
At entry	8.5 ± 2.0 (69.5 ± 22.4)
At last visit	8.0 ± 1.9 (63.4 ± 21.1)
Reduction since first assessment	0.6 ± 2.4 (29.5 ± 26.4)
T1DM HbA1c (%, mmol/mol)	
*n* (%)	57 (95.0)
Mean, mean ± SD	8.2 ± 1.3 (66.6 ± 14.6)
At entry	8.7 ± 2.0 (71.4 ± 21.3)
At last visit	8.1 ± 1.9 (64.7 ± 20.9)
Reduction since first assessment	0.6 ± 2.5 (30.2 ± 26.9)
Achieved national target ^6^, *n* (%)	21 (36.8)

Abbreviations: CSII = continuous subcutaneous insulin infusion (insulin pump), MDIs = multiple daily injections, CGM = continuous glucose monitor, TIR = time in range. ^1^ If treated with metformin at any point; ^2^ using a finger-prick glucometer; ^3^ as a percentage of patients with CGM; ^4^ number of days during reporting period with at least 50% CGM readings; ^5^ as a percentage of patients with uploaded data available; ^6^ as a percentage of T1DM patients.

## Data Availability

The raw, de-identified data supporting the conclusions of this article will be made available by the authors on request.

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
