# Peer review of "Diabetes Control and Clinical Outcomes among Children Attending a Regional Paediatric Diabetes Service in Australia"

_nutrients, 2024, doi:10.3390/nu16213779_

Round 1

Reviewer 1 Report

Comments and Suggestions for Authors

In the manuscript, "Diabetes Control and Clinical Outcomes Among Children Attending a Regional Pediatric Diabetes Service in Australia" by Huynh and colleagues. the authors have examined characteristics of children with diabetes over a 3 year period. While the authors have attempted to describe the cohort in detail, the manuscript only compares to some national guidelines and no comparison to other communities regionally or nationally. The following are comments and suggestions for the authors to consider:

1) The evaluation should be limited to T1D as that are the standards to which the population is currently being compared.

2) The authors comment that the children are not managed by a pediatric endocrinologist. A comparison to a similar group managed by a pediatric endocrinologist would be a very useful comparison.

3) The authors mention that they have used the International Obesity Task Force guidelines for obesity. These are mostly adult guidelines. The authors should be very clear what cut-points were used to define obesity. The authors might consider using the WHO guidelines.

4) The authors should attempt some correlation between mental disorder and A1c. Other correlations should also be considered. 

5) The authors should exam what characteristics predict or are associated with achieving national targets. Also consider the analysis of the characteristics associated with not meeting targets.

6) The discussion provides a review of characteristics and barriers that are associated with poor outcomes; however, it is not clear that these were included in the analysis or gathered as data in the study.

7) A minor comment, CGM is recommended by ADA standards of care.

Author Response

Response To Reviewers

Dear Dr [reviewer name],

Thank you for giving us the opportunity to submit a revised draft of the manuscript “Factors associated with loneliness in rural Australia: a web-based cross-sectional survey” for publication in the Journal of Nutrients. We appreciate the time and effort that you and the reviewers dedicated to providing feedback on our manuscript and are grateful for the insightful comments on, and valuable improvements to our paper. We have incorporated most of the suggestions made by the reviewers. Those changes are highlighted within the revised manuscript. Please see below, in blue, for a point-by-point response to the reviewers’ comments and concerns. All page numbers refer to the revised manuscript file. 

Reviewers; Comments and Authors’ Responses

Reviewer #1

In the manuscript, "Diabetes Control and Clinical Outcomes Among Children Attending a Regional Pediatric Diabetes Service in Australia" by Huynh and colleagues. the authors have examined characteristics of children with diabetes over a 3 year period. While the authors have attempted to describe the cohort in detail, the manuscript only compares to some national guidelines and no comparison to other communities regionally or nationally. The following are comments and suggestions for the authors to consider:

  1. Comment: The evaluation should be limited to T1D as that are the standards to which the population is currently being compared.

Author response: For the purposes of this audit, we wanted to report the characteristics of all patients who were seen at the diabetes clinic within this period for quality improvement. These included 57 patients with T1D, 2 with T2D, 1 person with MODY. However, when comparing to T1D treatment targets and national averages, we only included data from T1D patients. Other types of diabetes were not evaluated due to the small number of patients with those types. We acknowledge that this was not made clear in our discussion and have rectified this issue.  Please see the changes to the manuscript below.

Page 3 line 111-117:

This audit reported the characteristics of all patients seen at the diabetes clinic within this period for quality improvement. However, for comparing treatment targets with national averages, data for only T1D patients were used, due to the small number of people with other types of diabetes.

  1. Comment: The authors comment that the children are not managed by a pediatric endocrinologist. A comparison to a similar group managed by a pediatric endocrinologist would be a very useful comparison.

Author response: The authors agree that this would be a very useful comparison. Unfortunately, it is extremely rare for a regional/rural diabetes clinic to have care primarily provided by a paediatric endocrinologist. Many regional/rural clinics are run by general paediatricians or through telehealth or outreach models. Few clinics may occasionally consult with metropolitan paediatric endocrinologists via telehealth only. The lack of highly specialised care in regional/rural areas is an ongoing issue in Australia. Alternatively, we have included a paragraph comparing our study to a similar rural Australian study of a similar group which also involved a multidisciplinary model of care. Please see the changes to the manuscript below in green.

Page 5 line 176:

  • The mean HbA1c for T1DM patients (8.1 ± 1.4%) was similar (8.1 ± 1.4% versus 8.3 ± 3.5%; p = 0.326) to that reported in a large-scale multi-centre national audit (8.3 ± 3.5%) [7], an international study (8.2 ± 1.4%) [25], and a single-centre regional Australian study with a similar population (8.1 ± 1.3%) [16]; however nevertheless, it was higher than the national target of 7.5% (p = 0.001) which was achieved by only 36.8% of T1DM patients (n = 21/57) in this study.

Page 6 line 213:

  • … compared to those in the national audit (8% vs 27%) [7], a Queensland study (21%) [5], another large-scale, multi-centre Australasian study (27.2%) [6], and a UK study national audit (14.7%) [27]. Despite this, the mean HbA1c level (8.1%) did not differ significantly from national (8.3%) [7] or international (8.2%) standards averages [25], thus suggesting that, while the clinic has been relatively successful in helping individual patients, overall management still remains a challenge.

Page 7 line 222:

  • Our results align with findings from Goss et al. (2009), who implemented a multi-disciplinary model of care called ‘RADICAL’ in a similar population [16]. In addition to a significant reduction in mean HbA1c from 9.6% to 8.1%, their study reported improved quality of life for rural youth with T1DM, eliminating the gap previously seen between rural and urban diabetic populations. The diabetes clinic is already making good progress with a multidisciplinary approach, thus continuing to strengthen this model of care could further improve patient outcomes.

  1. Comment: The authors mention that they have used the International Obesity Task Force guidelines for obesity. These are mostly adult guidelines. The authors should be very clear what cut-points were used to define obesity. The authors might consider using the WHO guidelines.

Author response: As suggested by the reviewer, we have clarified the guidelines and cut-off points used to define overweight and obesity in our study. In Australia, WHO growth charts / criteria are used for children aged 0-2 years, and US-CDC for children aged 2-18 years. As all of the children in our study were older than 2 years, we have specified the use of the US-CDC cut-offs. There have been no changes to our results. Please see the changes to the manuscript and references below in green.

Page 3 line 99:

  • Anthropometry was taken at first and at last assessment; children were classified as nor-mal weight, overweight, or obese in accordance with International Obesity Task Force guidelines [20]. In Australia, WHO growth charts are used for children aged 0-2 years, while US-CDC growth charts are used for children aged 2-18 years [20, 21]. No patients in our study were younger than two years, thus they were all categorised according to the US-CDC cut-offs; a BMI-for-age between the 85th and 94th percentile was considered overweight, and above the 95th percentile was considered obese [23].

Page 5 line 158:

  • 1 defined according to International Obesity Task Force guidelines US-CDC growth charts

References:

  • Cole TJ. Establishing a standard definition for child overweight and obesity worldwide: international survey. BMJ. 2000;320(7244):1240.
  • Centre for Population Health. Growth Assessment in Children and Weight Status Assessment in Adults: NSW Government; 2017.
  • National Health and Medical Research Council. Clinical practice guidelines for the management of overweight and obesity in adults, adolescents and children in Australia: Melbourne: National Health and Medical Research Council; 2013.
  • Centers for Disease Control and Prevention. Child and Teen BMI Categories; 2024. Available from: https://www.cdc.gov/bmi/child-teen-calculator/bmi-categories.html.

  1. Comment: The authors should attempt some correlation between mental disorder and A1c. Other correlations should also be considered.

Author response: Thank you for this feedback. We have conducted statistical analyses to identify correlations between mental disorder and HbA1c, attendance rates, CGM usage, and hospitalisations. The results are as follows:

  • HbA1c (p = 0.059)
  • Attendance (p = 0.734)
  • CGM usage (p = 0.358)
  • Hospitalisations (p = 0.007)

Lack of association with some variables does not rule out the impact of mental health, but may be due to our small sample size or the limitations of using HbA1c as a metric. However, we found a significant association with number of hospitalisations. We have included all of these findings in our results and discussion. Please see the changes to the manuscript below in green.

Page 1 line 22:

  • Many patients had mental disorders (31.7%), and which was associated with higher hospitalisation rates (p = 0.007) had disproportionate numbers of hospital presentations.

Page 4 line 144:

  • Compared with people without mental health disorders, those with mental health disorders had higher HbA1c readings (7.9 ± 1.6% vs 8.6 ± 2.5%) which was almost significant (p = 0.059). Having a mental disorder was also not associated with attendance rates (p = 0.734) or CGM usage (p = 0.358).
  • There were 47 hospitalisations during the study period (four-year), involving over one-third of this cohort (21, 35.0%) including 16.7% with DKA, 18.3% with hyperglycemia, and 20.0% for elective admissions. Patients with mental disorders had significantly higher hospitalisation rates than those without (p = 0.007), and were disproportionately represented in the number of hospitalizations (61.7% of all presentations). The number of hospitalisations was also associated with higher mean HbA1c levels (r = 0.498, p < 0.001).

Page 6 line 207:

  • Mental disorders and diabetes-related hospitalisations were also common in this population. Those with a mental disorder were more likely to experience higher hospitalisation rates.

Page 7 line 268:

  • Current guidelines highlight the importance of psychosocial support during the transition to adult care [8], a sentiment reinforced by similar studies of transition clinics [11, 39]. Although mental disorder was only associated with some patient outcomes, this does not rule out the impact of mental health on diabetes care. A similar study of a transition clinic demonstrated significant associations between mental disorder and diabetes complications [11]. Our small sample size may have limited our ability to detect small changes and, while HbA1c is a key metric, it cannot provide a comprehensive picture of diabetes management for a patient. Nevertheless, the importance of providing psychosocial support for adolescents with diabetes is well established [8]. The mean age of those with mental disorders was 15.6 years at final assessment, reflecting the increased risk of psychiatric comorbidity associated with longer diabetes duration [40]. Despite the majority of patients being young adolescents, the current MDT lacks a social worker or psychologist; according to the ISPAD guidelines, there should be 0.3 FTE social worker/psychologist per 100 patients [36].

  1. Comment: The authors should exam what characteristics predict or are associated with achieving national targets. Also consider the analysis of the characteristics associated with not meeting targets.

Author response: The authors agree that examining these characteristics would be a very useful contribution to the literature. However, we believe that a thorough examination is beyond the scope of our study, which primarily aims to characterise the patients attending a regional paediatric diabetes clinic for the purposes of quality improvement. Furthermore, the small sample size of our study would limit the statistical power of any conclusions drawn from the analyses. The significant associations we have identified, such as between HbA1c and hospitalisations, CGM sensor usage and TIR, attendance, and HbA1c, as well as comparing mean HbA1c to other study populations, already provide valuable insight into T1DM management in regional/rural communities. Nevertheless, we appreciate the reviewer’s comment and will try to address this point in our discussion and in our recommendations for future research. Please see the changes to the manuscript and references below in green.

Page 8 line 291:

  • This These results aligns with findings from a recent study of 25,383 children with T1DM, which identified CGM use as a major factor in meeting HbA1c targets highlighting the rapid uptake of CGM following the introduction of government subsidies (5% to 79% after 2 years) and its positive impact on diabetes control [42].

Page 8 line 309:

  • A longer study time frame, multiple centres, and broader scope of variables (complications, screening, metabolic outcomes) would allow for enable a more detailed evaluations of outcomes and trends examination of factors associated with achieving target outcomes.

References:

  • Johnson SR, Holmes-Walker DJ, Chee M, Earnest A, Jones TW. Universal Subsidized Continuous Glucose Monitoring Funding for Young People With Type 1 Diabetes: Uptake and Outcomes Over 2 Years, a Population-Based Study. Diabetes Care. 2022;45(2):391-7.
  • Demeterco-Berggren C, Ebekozien O, Noor N, Rompicherla S, Majidi S, Jones N-HY, et al. Factors Associated With Achieving Target A1C in Children and Adolescents With Type 1 Diabetes: Findings From the T1D Exchange Quality Improvement Collaborative. Clinical Diabetes. 2023;41(1):68-75.

  1. Comment: The discussion provides a review of characteristics and barriers that are associated with poor outcomes; however, it is not clear that these were included in the analysis or gathered as data in the study.

Author response: Reasons for missed appointments were mostly documented in the electronic medical records (eMR), and we created a list of recurrent reasons whilst collecting the data. This was reported on page 3 line 125. However, we agree with the reviewer that this was not made very clear in our report – in particular, we did not include “reasons for missed appointments” as a type of data collected within our materials and methods section. We have also changed the phrasing of this part of our discussion to avoid any potential misunderstandings. Please see the changes to the manuscript below in green.

Page 2 line 87:

  • Types of data collected included patient information, demographics, attendance, reasons for missed appointments, anthropometry, clinical profile, treatment modalities, clinical outcome measures, and reasons for leaving the clinic.

Page 7 line 243:

  • Lack of transportation, was a common one of the causes of missed appointments, and is known to be a major barrier to healthcare access in regional Australia [18, 30].

  1. Comment: A minor comment, CGM is recommended by ADA standards of care.

Author response: Thank you for this feedback. We have now clarified in our introduction that, although CGM use is not routinely recommended in Australia, it is indeed recommended by the ADA standards of care. Please see the changes to the manuscript and references below in green.

Page 1 line 45:

  • One such technology is continuous glucose monitoring (CGM); it is not currently recommended for routine use, so, with only few Australian studies have having examined its role use within T1DM clinics [11,12]. Although the American Diabetes Association (ADA) recommends that CGM be offered as soon as possible [13], Australian guidelines currently do not advocate for its routine use [8].

References:

  • Elsayed NA, Aleppo , Bannuru RR, Bruemmer D, Collins BS, Ekhlaspour L, et al. 7. Diabetes Technology: Standards of Care in Diabetes—2024. Diabetes Care. 2024;47(Supplement_1):S126-S44.

Reviewer 2 Report

Comments and Suggestions for Authors

The manuscript by Luke Huynh et al. investigated the diabetes control and clinical outcomes among children attending a regional paediatric diabetes service in Australia. The study is well-designed and the findings are interesting. I have some suggestions:

1, why so many children have type 1 diabetes in Australia? Any explanations? 

2, the sample size of the present study is very small. Could this be improved? 

3, why so many patients had mental disorders? Any explanations? 

Author Response

Response To Reviewers

Dear Reviewer

Thank you for giving us the opportunity to submit a revised draft of the manuscript “Factors associated with loneliness in rural Australia: a web-based cross-sectional survey” for publication in the Journal of Nutrients. We appreciate the time and effort that you and the reviewers dedicated to providing feedback on our manuscript and are grateful for the insightful comments on, and valuable improvements to our paper. We have incorporated most of the suggestions made by the reviewers. Those changes are highlighted within the revised manuscript. Please see below, in blue, for a point-by-point response to the reviewers’ comments and concerns. All page numbers refer to the revised manuscript file. 

Reviewer #2

The manuscript by Luke Huynh et al. investigated the diabetes control and clinical outcomes among children attending a regional paediatric diabetes service in Australia. The study is well-designed and the findings are interesting. I have some suggestions:

  1. Comment: Why so many children have type 1 diabetes in Australia? Any explanations?

Author response: The prevalence of type 1 diabetes in Australian children (22 per 100,000 person-years) is considered very high compared to other countries.

  • Catanzariti, L. et al. (2009) ‘Australia’s national trends in the incidence of type 1 diabetes in 0–14yearOlds, 2000–2006’, Diabetic Medicine, 26(6), pp. 596–601. doi:10.1111/j.1464-5491.2009.02737.x.

Various hypotheses have been presented for why this is the case, many revolving around changes in environmental factors in modern society. Some of the factors suggested include: delivery method, early lifestyle factors, microbe exposure / hygiene hypothesis, pollution, gut biome, and natural selection. Below are some interesting papers which shed light on this complex question.

  • Ogrotis, I., Koufakis, T. and Kotsa, K. (2023) ‘Changes in the global epidemiology of type 1 diabetes in an evolving landscape of environmental factors: Causes, challenges, and opportunities’, Medicina, 59(4), p. 668. doi:10.3390/medicina59040668.
  • You W, Henneberg M. Type 1 diabetes prevalence increasing globally and regionally: the role of natural selection and life expectancy at birth. BMJ Open Diabetes Research and Care 2016;4:e000161. doi: 10.1136/bmjdrc-2015-000161
  • Gale, E.A.M. (2002) ‘The rise of childhood type 1 diabetes in the 20th century’, Diabetes, 51(12), pp. 3353–3361. doi:10.2337/diabetes.51.12.3353.

  1. Comment: The sample size of the present study is very small. Could this be improved?

Author response: The authors agree that the sample size of this study is small since it is a single-centre study. However, we studied the total population of children attending the clinic within the four-year time frame, allowing us to reliably evaluate the clinic. We have acknowledged the small sample size as a limitation of our study in our discussion on page 8 line 300.

  1. Comment: Why so many patients had mental disorders? Any explanations?

Author response: The link between mental disorder and diabetes is well established. However, the reasons behind this are complex and are an interesting area of future research. Below are some articles of interest:

  • Akhaury, K., & Chaware, S. (2022). Relation Between Diabetes and Psychiatric Disorders. Cureus, 14(10), e30733. https://doi.org/10.7759/cureus.30733]
  • Xie, X.-N. et al. (2022) ‘Association between type 1 diabetes and neurodevelopmental disorders in children and adolescents: A systematic review and meta-analysis’, Frontiers in Psychiatry, 13. doi:10.3389/fpsyt.2022.982696.

The number of patients with a mental disorder in our study population was comparable to the number in a similar study of a transition clinic (31.7% vs 40.5%):

  • Sritharan A, Osuagwu UL, Ratnaweera M, Simmons D. Eight-Year Retrospective Study of Young Adults in a Diabetes Transition Clinic. International Journal of Environmental Research and Public Health. 2021;18(23):12667

We have performed further statistical analyses to investigate associations between mental disorder and clinical outcomes such as HbA1c, attendance rates, CGM usage, and hospitalisations. Please see our response to Reviewer #1 Comment 4 above.

Reviewer 3 Report

Comments and Suggestions for Authors

Review for the manuscript Diabetes Control and Clinical Outcomes Among Children Attending a Regional Paediatric Diabetes Service in Australia

The present study evaluated children attending Bathurst's diabetes service in order to improve diabetes management in this region of Australia. Most of the patients had type 1 diabetes, the treatment consisting in subcutaneous insulin infusion (CSII) pumps, or multiple daily injections (MDI). The authors evaluated attendance rates, anthropometry, clinical history, and treatment outcomes. Very interesting results for clinicians as well for patients.

The authors said that 31.7% of patients had mental disorders. Do the authors think that the results of the study are influenced by this fact?

The authors should mention the criteria used to classify children as normal weight, overweight, or obese. Maybe they could present the results of the biochemical analysed performed.

Line 161: HbA1c levels were available for all but 1 patient (98.3%); what does it mean?

Maybe it could be interesting if the authors analysed other metabolic parameters in association with HbA1c levels.

I consider that the results can be improve.

Author Response

Response To Reviewers

Dear Dr [reviewer name],

Thank you for giving us the opportunity to submit a revised draft of the manuscript “Factors associated with loneliness in rural Australia: a web-based cross-sectional survey” for publication in the Journal of Nutrients. We appreciate the time and effort that you and the reviewers dedicated to providing feedback on our manuscript and are grateful for the insightful comments on, and valuable improvements to our paper. We have incorporated most of the suggestions made by the reviewers. Those changes are highlighted within the revised manuscript. Please see below, in blue, for a point-by-point response to the reviewers’ comments and concerns. All page numbers refer to the revised manuscript file. 

Reviewers; Comments and Authors’ Responses

Reviewer #3

Review for the manuscript Diabetes Control and Clinical Outcomes Among Children Attending a Regional Paediatric Diabetes Service in Australia
The present study evaluated children attending Bathurst's diabetes service in order to improve diabetes management in this region of Australia. Most of the patients had type 1 diabetes, the treatment consisting in subcutaneous insulin infusion (CSII) pumps, or multiple daily injections (MDI). The authors evaluated attendance rates, anthropometry, clinical history, and treatment outcomes. Very interesting results for clinicians as well for patients.

  1. Comment: The authors said that 31.7% of patients had mental disorders. Do the authors think that the results of the study are influenced by this fact?

Author response: Thank you for your comment. The link between mental disorder and diabetes is well established in the literature. While mental health was not the primary focus of the study, it is a variable that could impact adherence to treatment and diabetes control. Although we were unable to draw strong conclusions regarding mental disorder and patient outcomes, other studies of similar population groups have demonstrated the negative impact of mental disorder on diabetes control. Please see the above responses to Reviewer #1 Comment 4 and Reviewer #2 Comment 3 for more detailed analyses regarding this topic. Future studies could further explore this relationship to better understand the effects of mental health on diabetes outcomes

  1. Comment: The authors should mention the criteria used to classify children as normal weight, overweight, or obese. Maybe they could present the results of the biochemical analysed performed.

Author response: Please see the above response to Reviewer #1 Comment 3.

  1. Comment: Line 161: HbA1c levels were available for all but 1 patient (98.3%); what does it mean?

Author response: We note that this sentence could be better worded to avoid any potential misunderstandings. Please see the changes to the manuscript below in green.

Page 5 line 174:

  • HbA1c levels were data was available for all but 1 patients (3%) of patients

  1. Comment: Maybe it could be interesting if the authors analysed other metabolic parameters in association with HbA1c levels.

Author response: Thank you for your suggestion. The authors agree that analysing other metabolic parameters would be quite interesting. Unfortunately, the clinic in our study did not routinely measure other parameters such as lipids, creatinine, albumin, etc. for patients. We acknowledge that future studies should ideally include more metabolic outcomes in our discussion on page 8 line 309.

I consider that the results can be improve.

Round 2

Reviewer 1 Report

Comments and Suggestions for Authors

The authors have made significant improvements to the clarity of the manuscript; however the following concerns still exist:

1) The authors mention that change in HbA1c overtime was last-first A1c. A more accurate reflection of A1c over time would be to calculate an area under the curve or a time weighted average.

2) Depression and anxiety should be classified as mood disorders which are very different from ADHD. These should be separated and analyzed independently. 

3) More clarity is needed with regard to the elective admissions. For the correlation with A1c, the admissions should be separated into those that are diabetes related and those that are not.

4) The take away message seems to be that the clinic is doing as well as others and no one is meeting the treatment targets. This is not conveyed clearly in the manuscript. 

5) The abstract alludes to improvements in clinic infrastructure, but these are not detailed in the manuscript (or are not clearly communicated). It is not clear if there was any analysis before or after these changes.

Author Response

The authors have made significant improvements to the clarity of the manuscript; however the following concerns still exist:

  1. Comment: The authors mention that change in HbA1c overtime was last-first A1c. A more accurate reflection of A1c over time would be to calculate an area under the curve or a time weighted average.

Author response: The clinic in our study consistently measured HbA1c levels every clinic visit (i.e. every 3 months) for each patient. We collected this data and recorded the initial, final, change (final minus initial), and mean HbA1c for each patient. We then calculated the mean value for all patients for each of these parameters – i.e. mean initial, mean final, mean change, mean (which is the overall mean for the 4-year study period). The authors agree that last-first HbA1c does not capture fluctuations in A1c in between these measurements. However, we did not intend for this to be any more than a rough snapshot of patients at first and last appointments for the clinic audit. When comparing glycaemic control with other studies or with national standards, similarly to many other studies, we used the overall mean HbA1c as a more accurate reflection of A1c.This also happens to be a time-weighted average since the measurements were taken at regular intervals of 3 months.

  1. Comment: Depression and anxiety should be classified as mood disorders which are very different from ADHD. These should be separated and analysed independently.

Author response: Thank you for your comment. The authors agree that mood disorders are different from ADHD, which is a neurodevelopmental disorder. According to the WHO, ‘mental health condition’ is a broader term covering mental disorders, psychosocial disabilities and (other) mental states associated with significant distress, impairment in functioning, or risk of self-harm. We believe this term more accurately represents the conditions reported in our study as a whole, and will change this in our manuscript. At the beginning, we decided to group these conditions together in order to more broadly examine the effects of mental health conditions on diabetes control. Although it is certainly possible to separate the different types of disorders and evaluate them individually, the smaller sample size would make it very difficult to draw any nuanced conclusions. Based on the reviewer’s feedback, we will note this as a point of improvement in our discussion.

  • Mental disorders (2022) World Health Organization. Available at: https://www.who.int/news-room/fact-sheets/detail/mental-disorders#:~:text=Mental%20disorders%20may%20also%20be,or%20risk%20of%20self%2DDhar.

‘Mental disorder’ was changed to ‘mental health conditions’, or likewise, in the following locations:

  • Abstract
  • Page 4 line 150, 153, 155
  • Table 2
  • Page 7 line 217, 219
  • Page 8 line 281, 288
  • Page 9 line 334

Page 9 line 319 (discussion):

  • Mental health conditions could be further categorised for a more nuanced understanding of its impact on diabetes control.

  1. Comment: More clarity is needed with regard to the elective admissions. For the correlation with A1c, the admissions should be separated into those that are diabetes related and those that are not.

Author response: Thank you for this feedback. We realised we have not been clear in defining this in our report. “Elective admissions” were when the treating doctor in the MDT admitted a patient to hospital due to concern regarding diabetes control. The goals of admission could be either for observation/monitoring, or to optimise treatment regimes. We will clarify this point in our report, in particular changing the term to “planned admissions”. Please see the changes to the manuscript below in green.

Page 2 line 97:

  • Clinical profile included anthropometry, substance use, diabetes-related hospitalisations (emergency or admissions), comorbidities, and mental health conditions.

Table 2:

  • Elective Planned admission
  • Multiple elective planned admissions

Page 5 line 170:

  • 5 for monitoring or for optimisation of treatment.

  1. Comment: The take away message seems to be that the clinic is doing as well as others and no one is meeting the treatment targets. This is not conveyed clearly in the manuscript.

Author response: The clinic performance is similar to that reported in other studies with regards to glycaemic control, with 36.8% of patients meeting treatment targets. Mental health and overweight/obesity have been identified as ongoing issues. However, the clinic performs well with regards to attendance and uptake of diabetes technology. Thank you for your feedback. We have now clarified these points in our abstract.

Abstract:

  • Recent large-scale audits have revealed poor levels of diabetes control among Australian children with diabetes often struggle to achieve optimal glycaemic control, with minimal improvement observed over the past decade.
  • This study aimed to characterise the children attending a regional paediatric diabetes service, evaluating various clinical outcomes to assess the effectiveness of treatment protocols. We conducted a A retrospective audit was conducted of all 60 children who attended the attending a regional Australian paediatric diabetes service clinic between January 2020 and December 2023.
  • Almost all The majority of patients had type 1 diabetes (n = 57, 95.0%)
  • Despite notable improvements in clinic infrastructure Overall, the diabetes service performed similarly to other clinics with regards to glycaemic control. achieving optimal glycaemic control and Whilst achieving treatment targets and addressing associated comorbidities remains a challenge, the high uptake of diabetes technologies is commendable. emphasising the need for continued Further efforts are needed to enhance improve diabetes management in for this regional community.

  1. Comment: The abstract alludes to improvements in clinic infrastructure, but these are not detailed in the manuscript (or are not clearly communicated). It is not clear if there was any analysis before or after these changes.

Author response: Thank you for this feedback. The word “improvements” should not have been used as these changes have not been evaluated yet. We have amended our terminology in the revised manuscript. Regarding this point, we were alluding to various changes to the model of care described in ‘Study Setting’ on page 2 line 67. As these changes occurred at different points in time, we determined it would be difficult to evaluate the clinic before and after them. Thus, our study focused on performing a general audit of the clinic for the purposes of quality improvement rather than evaluating the impact of changes. Nevertheless, the authors agree that our descriptions of these changes in other parts of the manuscript are unclear, and have revised accordingly. We also highlighted in our discussion that we were unable to determine the effectiveness of these changes for the above reasons. Please see the changes to the manuscript below in green.

Page 1 line 27:

  • Despite notable improvements in clinic infrastructure Overall, the diabetes service performed similarly to other clinics with regards to glycaemic control. achieving optimal glycaemic control and Whilst achieving treatment targets and addressing associated comorbidities remains a challenge, the high uptake of diabetes technologies is commendable. emphasising the need for continued Further efforts are needed to enhance improve diabetes management in for this regional community.

Page 8 line 313:

  • We also did not account for differences in appointment durations between paediatricians, the effects of changes to the model of care, or the impact of socioeconomic factors.

Conclusion:

  • Despite notable improvements in clinic infrastructure various changes to the model care, achieving optimal glycaemic control and addressing associated comorbidities remains a challenge.

Reviewer 2 Report

Comments and Suggestions for Authors

The authors have revised the manuscript accordingly. It can be considered for publication. 

Author Response

(The authors gave the same response as above.)
